# Novel method to investigate thermal exchange rates in small, terrestrial ectotherms: A proof-of-concept on the gecko *Tarentola mauritanica*

**Gabriel Mochales-Riaño**[1,2☯], **Frederico M. Barroso**[1,3,4☯]*, **Valéria Marques**[1,2,3], **Alexandra E. Telea**[1,5], **Marco Sannolo**[6], **Catarina Rato**[1,4], **Miguel A. Carretero**[1,3,4]

**1** CIBIO, Research Centre in Biodiversity and Genetic Resources, InBIO, Universidade do Porto, Vila do Conde, Portugal, **2** Institute of Evolutionary Biology (CSIC-Universitat Pompeu Fabra), Barcelona, Spain, **3** Faculdade de Ciências, Departamento de Biologia, Universidade do Porto, Porto, Portugal, **4** BIOPOLIS Program in Genomics, Biodiversity and Land Planning, CIBIO, Vairão, Portugal, **5** Faculty of Natural and Agricultural Sciences, Ovidius University of Constanța, Constanța, Romania, **6** Museo Nacional de Ciencias Naturales (MNCN)–CSIC, Madrid, España

☯ These authors contributed equally to this work.
* frederico.m.barroso@gmail.com

**Data Availability Statement:** All data and R scripts for statistical analyses are openly available in the

## Abstract

Thermoregulating ectotherms may resort to different external heat sources to modulate their body temperature through an array of behavioural and physiological adaptations which modulate heat exchange with the environment and its distribution across the animal's body. Even small-bodied animals are capable of fine control over such rates and the subsequent re-allocation of heat across the body. Such thermal exchanges with the environment usually happen through two non-mutually exclusive modes: heliothermy (radiant heat gain from the sun) or thigmothermy (heat gained or lost via conduction). Classically, the study of these phenomena has relied on invasive methodologies which often disregard the effect of stress, behaviour and regional heterothermy on the rates and patterns of thermal exchange across the body of the animal. This study proposes a novel experimental methodology, capitalising on thermography, to provide an alternative method to less invasively obtain reliable body temperatures of thermoregulating ectotherms, while allowing behaviour and heating mode to be considered when quantifying thermal exchange rates. This methodology was tested in the gecko *Tarentola mauritanica*, where twenty males were allowed to heat up and cool down under a novel experimental set-up which isolates heliothermic and thigmothermic processes, while being recorded with a thermal camera. The study revealed differences in the heating and cooling rates of several body parts per treatment suggesting that thermal exchanges are complex even in small ectotherms. Ultimately, the described set-up provides the opportunity to revisit classical questions with a less invasive and more flexible experimental approach, enabling heliothermic and thigmothermic processes to be disentangled. The described methodology also better integrates behaviour and physiology while obtaining higher temporal and spatial resolution of body temperatures in a thermoregulating ectotherm.

following GitHub repository: https://github.com/FredericoMB/Tarentola_Thermal_Exchange_Rates.

**Funding:** This study was carried out under the project PTDC/BIA-REP/27958/2017 awarded to CR, funded by FCT- Fundação para a Ciência e a Tecnologia (https://www.fct.pt/), Portugal. MS was supported by the project PTDC/BIA-CBI/28014/2017; CR by a FCT postdoctoral contract (DL57/2016/CP1440/CT0005), GM-R and VM by the FCT project PTDC/BIA-REP/27958/2017 as well as by an FPI grant from the Ministerio de Ciencia, Innovación y Universidades (https://www.ciencia.gob.es/), Spain (PRE2019-088729 for GM-R and PRE2020-094870 for VM); FMB by FCT PhD grants (SFRH/BD/147535/2019 and COVID/BD/153468/2023); and AET by an Erasmus+ (https://erasmus-plus.ec.europa.eu/) grant (1215/29.01.2020). Work supported by the European Union's Horizon 2020 Research and Innovation Programme (https://research-and-innovation.ec.europa.eu/) under the Grant Agreement Number 857251. The funders had no role in study design, data collection and analysis, decision to publish, or preparation of the manuscript.

**Competing interests:** The authors have declared that no competing interests exist.

## Introduction

Thermoregulating ectotherms, such as many reptiles, must rely on a combination of external heat sources, behavioural strategies and physiological mechanisms to modulate body temperature within an optimal range [1–3]. This approach allows reptiles to occupy a wide range of environments, from tropical forests and deserts to temperate regions and even subarctic areas [4,5]. Compared to other vertebrates, reptiles highlight the role of behavioural thermoregulation for maintaining, and even increasing, a species distribution and diversity from what is expected given their physiological critical limits [6–8]. Indeed, behavioural thermoregulation in reptiles has been recognized as a pivotal element in their biology, ecology and physiology (e.g. [9,10]), allowing them to attain body temperatures close (or equal) to the thermal optima maintaining a range of physiological processes, without the costly energetic needs associated to endothermy [11,12].

Heat exchange between the environment and an organism has been deeply studied since the 1950's aiming to understand, often from a theoretical point of view, the thermodynamic processes governing the mechanistic interplay between an animal's thermal state and its surroundings (e.g. [13,14]). In essence, these thermal exchanges occur through a combination of non-mutually exclusive radiative, convective, conductive, and evaporative processes [14,15] (explained by Birkebak [16]) on which a thermoregulating animal may exert some control, via physiology and/or behaviour, to optimise its thermal exchanges.

For ectotherms in temperate regions, convection is expected to provide a lesser contribution towards the increase in body temperature yet, along with evaporation [17,18], may be an important mechanism of heat loss when an organism operates close to the upper limits of their thermal range [19–22]. Moreover, dehydration tends to restrict temperate reptiles to body temperatures lower than their optima, when water is not available for evaporative cooling mechanisms to be engaged [23].

For cooling, however, and particularly in cold or windy environments, convection usually underpins an inherent, and often counter-productive [24,25] (but see works from Gontijo et al. [26]), loss of body temperature [27,28], which may represent significant costs to the temperate ectotherm. These costs may include restricted activity periods, increased time spent thermoregulating and increased risk of dehydration [24,25,29]. Hence, ectotherms from such environments often show a range of behavioural and physiological adaptations to minimise the rates of disadvantageous heat loss from convection and maximise the rates of heat gain, usually from radiation and/or conduction [30,31].

Finally, radiation and conduction are the two most effective and ubiquitous ways that reptiles utilise to modulate their body temperature towards a preferred set-point [32]. The active process of regulating body temperature from solar radiation (i.e. heliothermy) is a prominent strategy used to warm up and has allowed high levels of diversification in several reptile groups from a range of environments (e.g. [33]). On the other hand, the active exchange of heat through direct contact with the substrate (i.e. thigmothermy) is less ubiquitously reported (but see works from Ortega et al. [34,35]) and has frequently been associated with species with limited access to solar radiation, such as nocturnal, forest or fossorial animals [32]. However, it may also represent a complement to heliothermy, even in primarily heliothermic species [36]. Notably, heliothermy (i.e. heating up by IR radiation from the sun) is usually a unidirectional process since the initial heat supply will only act as a source but not as a sink. Nonetheless, it is relevant to point out that any hot body–including the animal–becomes a radiant heat source itself, as it inevitably emits IR radiation to its surroundings, hence effectively cooling down by means of radiation [37,38].

Thigmothermy, however, is inherently a more bidirectional phenomenon whereby the direction and rate of heat transfer depend respectively on the direction and magnitude of the

temperature differential between the animal and its substrate [2]. Hence, the animal can actively select a substrate either warmer or colder than itself to act as a heat source or sink. Thus, thigmothermy can aid in the fine tuning of an animal's body temperature and its distribution across the body, even while heating heliothermically [2]. This suggests that, although less studied experimentally (see works from Ortega et al. [34,35]), thigmothermy may be more common than often considered. For example, it is the main, but not exclusive, thermoregulatory strategy in the highly diverse group of Gekkota [32].

Ectotherm thermal ecology studies abound with a range of tools and methods used to acquire body temperature of the animals, both in field and laboratory contexts—extensively reviewed by Camacho & Rusch [39], and Taylor et al. [4]. In this field, there is an increasing trend to use more powerful but less invasive temperature measurement tools such as infrared (IR) thermometers (e.g. [40,41]), dataloggers (e.g. [29]) and thermal cameras (e.g. [42–44]). This, in turn, is also allowing for more integrative methodologies (e.g. [28,45]. Notwithstanding, although such tools are increasingly common in research on ectotherm thermal preference, tolerance or performance, they are still uncommon in studies evaluating thermal exchange rates of these animals (but see work from Rutschmann et al.[46]).

Historically, many of such studies on thermal exchange rates involved restraining the animal under a heat source while measuring its body temperature change at given time intervals, usually via a cloacal temperature probe (e.g. [47–51]). These focused on the physiological control of heating rates, disregarding the effect of behaviour (such as posturing), and imposing high stress levels, while neglecting differential heat distribution patterns to body parts other than the cloaca. Only a few studies opted for less manipulative approaches where animals were allowed to freely thermoregulate in a terrarium with only the desired heating mode (heliothermic or thigmothermic heat source) available throughout (e.g. [2]). While the latter scheme allowed for more natural thermoregulatory behaviour and physiology, this less invasive approach inevitably came at the cost of measuring temperature of the animals less often as well as in a single body part (i.e. the cloaca). Hence, studies addressing this trade-off and allowing to measure temperature less-invasively, continuously and in multiple body parts, may extend such findings to spatial and temporal patterns of heat gain.

Meanwhile, the use of thermography has been rising in thermal ecology and physiology studies [28,52,53]. Often, radiometric photos of a specific point in time are used to measure the body temperature of one or more body parts (e.g. [43,44]). This constitutes an improvement on previous methodologies (but see works from Zhang et al. [54]), which either relied on more invasive techniques (e.g. cloacal measurements) [47–51]) or failed at attaining a short-term continuum of measurements [2]. Nevertheless, thermography-based studies have so far also mostly disregarded the integration of behavioural data and continuous, non-invasive temperature measurements (but see works from Hastings et al. [45] and Rutschmann et al. [46]). Ultimately, this hindered the monitoring of fast physiological processes, such as maximum rates of thermal transfer and dynamic patterns of regional heterothermy, under behaviourally thermoregulating animals.

Here we provide and implement an alternative methodology that relies on radiometric video to investigate fast thermal exchange processes in small lizards, with the main aim of acquiring localised heat exchange rates of different body parts. Furthermore, we present and test a novel and flexible set-up to heat up or cool down an animal while controlling, or isolating, for the mode of heat exchange (heliothermic vs thigmothermic). We selected the crepuscular gecko *Tarentola mauritanica* (Linnaeus, 1758) as a model species due to its behavioural and physiological adaptations for obtaining heat through both thigmothermy and heliothermy. Ultimately, this method allows exploring differences in heat exchange patterns of different body parts, depending on the thermoregulatory strategy of interest (heliothermy vs

thigmothermy), while accounting for both physiological and behavioural variables, such as the ability to regulate blood flow across the body and to adjust body posturing, respectively.

## Materials and methods

### Study species

The common wall gecko, *Tarentola mauritanica* (Linnaeus, 1758), is a mid-sized (mean SVL = 70mm, mean mass = 9g) temperate lizard species with a circummediterranean distribution [55]. Although it can be found basking, capitalising on both heliothermy and thigmothermy, especially during the morning or late afternoon [56], most of its activity occurs at night when it is expected to use the thermal inertia of its substrate to warm up thigmothermically. This bimodal thermoregulation pattern makes this species an interesting model to study the physiology of thermal exchange rates.

### Sampling and husbandry

Animals were collected during spring 2019 from four different populations: Évora (coordinates: 38.532, -8.018), Torres Vedras (39.087, -9.246) and Portimão (37.197, -8.539) in Portugal, and Ayamonte (37.260, -7.347) in Spain. The geckos were attracted by a laser pointer on vertical surfaces [57] and then caught using a slipknot [58] or by hand. The animals were housed in 50Lx30Wx25H cm terraria with stacked rocks as refugia and a water dish for hydration. Kept in an air-conditioned room at 19°C, animals were provided heat from dusk until dawn (19:00h to 07:00h) through an array of 150W infrared light bulbs. Natural photoperiod was maintained through a diffuse glass window. Food consisting of house crickets (*Acheta domesticus*) or mealworm larvae (*Tenebrio molitor*) was provided three times a week.

Only adult males were used in the experiments to prevent any possible effects of ontogeny or of oogenesis (in females). To sex the animals, the base of the tail, adjacent to the cloaca, was gently squeezed to evert the hemipenes of males [59]. Five individuals per population were randomly chosen, totalling 20 animals with different body sizes. Since multiple populations were sampled and the size of *T. mauritanica* differs considerably between populations [60], the resulting sample encompassed a typical overall size range for adult males of this species. This aided in accounting for the potential effect of size (and hence thermal inertia) in the rates of thermal exchange and patterns of heat distribution across the body. For each animal, snout-to-vent length (SVL) was measured to the nearest 0.01mm using digital callipers (Perel Digital Calliper 3472, precision: 0.01mm, accuracy: ±0.02mm) and mass with a precision balance (Sartorius M-Pact AX224, Sartorius AG, Goettingen, Germany, precision: 0.0001g). Mean SVL ± standard error (SE) for the tested individuals was 70.01 ± 1.76mm (SVL range: 55.21–82.17mm; S1 Fig) and mean mass ± SE was 8.87 ± 0.46g during the heliothermic trials, and 8.90 ± 0.45g during the thigmothermic trials (overall mass range: 6.37–12.09g; S1 Fig).

To account for any individual-related discrepancies in the physiological responses, each individual was subjected to the two trials, in a random order, where different heating strategies (heliothermy vs thigmothermy), and the subsequent cooling process, were independently tested.

### Experimental settings

In order to isolate heliothermy from thigmothermy, independent experimental arenas were devised. The set-up for simulating heliothermy (for simplicity, hereafter referred to as "heliothermy experiment") consisted of a 110Lx50Wx30H cm aquarium partially filled with water, inside which a smaller (34Lx23Wx20H cm) aquarium was placed inside it, upside down

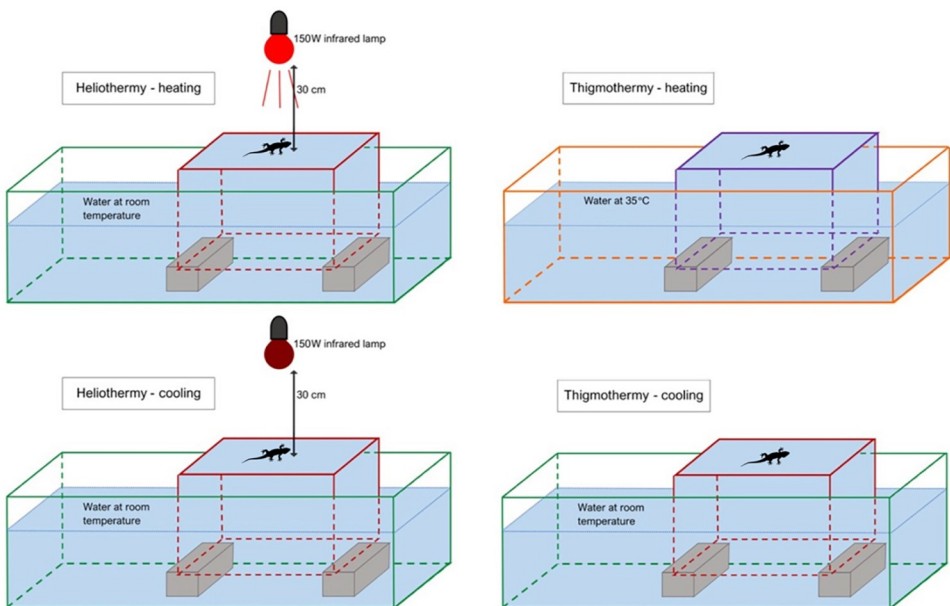

**Fig 1. Layout of the setup utilised for both heliothermy and thigmothermy experiments.** An inverted, smaller aquarium is placed inside a larger one and filled with water by vacuum suspension. On the top, both different setups are in heating mode and at the bottom they are in cooling mode. Lizard silhouette marks the spot (external glass surface of the inverted, smaller aquaria) where the gecko was positioned during the experiment. Blue shading represents the water-filled volumes of the aquaria.

and partially (5-10cm) emerging above the larger aquarium's water level. The small, inverted aquarium was also filled with water, through vacuum suspension, resulting in a combined volume of water of circa 170 litres of water (Fig 1). The bottom external face of the small aquarium's surface served as the experimental arena on which the animals were positioned. Only glass aquariums were used, and the thickness of the glass wall kept constant (4mm), in order to ensure similar heat conductivity of the arena's surface. A 150W infrared (IR) bulb was suspended 30cm above the arena's surface. The water in both aquaria was monitored through a contact thermometer (Hibok 14, precision: 0.1˚C, accuracy: 0.3%) fitted with a k-type thermocouple probe inserted inside the small aquarium. This water was kept at 19.0 ± 0.5˚C by the thermal inertia of the large volume of the system and through periodic changes or top-ups with hot or cold water as needed. To avoid thermal gradients forming in the water column, pumps were used for circulation and mixing (total flow rate > 6500 l/h). During the heating part of the experiment, the IR bulb was kept on and then subsequently turned off for the cooling part of the trial (Fig 1).

For the thigmothermy experiment, two separate experimental arenas were set up: one for the heating trial and another for the subsequent cooling trial. For heating, a setup similar to the one used for the heliothermy experiment (Fig 1) was used. However, the water was previously heated and kept at 35.0 ± 0.5˚C using aquarium heaters (EHEIM Thermocontrol 100W, adjustable temperature range: 18–34˚C, Accuracy: 0.5˚C, EHEIM GmbH & Co, KG, Germany), heat matts attached to the outer glass surface of the main tank and occasional top-ups with heated water. As in the heliothermic set-up, water temperature was monitored through the same k-type thermocouple probe inside the internal and inverted smaller aquarium and, as before, temperature gradients were avoided by the use of submerged pumps. The large volume of water and subsequent large thermal inertia facilitated the maintenance of a stable temperature.

For the cooling part of the thigmothermic experiment, immediately after the end of the heating trial, the animal was moved to a smaller replica of the experimental arena (a 60Lx30Wx28H cm aquarium with a 30Lx18Wx20H cm aquarium turned over inside resulting in a combined volume of water of circa 60 litres), with water at 19 ± 0.5˚C (Fig 1)—maintained as described for the heliothermic set-up. The differences in the dimensions of the aquaria used were solely due to availability of these. Nevertheless, future deployments of this methodology should aim to set-up exact replicas (i.e. same size, same total volume of water) of the arenas for both the heating and cooling trials.

The relocation from the thigmothermic heating arena to the respective cooling arena was fast (~2 seconds) by using a harness and leash, previously fitted to the animals to avoid touching them during the trial, which would interfere with their body temperature. Upon contact with the cooling arena, the video recordings and necessary additional measurements were immediately initiated synchronously (explained below).

For all the experiments, room temperature was controlled via an air conditioning unit set to 19.0 ± 0.5˚C and measured with a thermometer (Fluke 971 Temperature and Humidity Meter, precision: 0.1˚C, accuracy: ± 0.5˚C). Overall, air and water temperature were constant along the experiments, with mean air temperature of 18.83 ± 0.34˚C, and water temperature of 18.82 ± 0.68˚C for all the cooling and the heliothermic heating trials and 35.13 ± 0.26˚C for the thigmothermic heating trials.

Prior to the experiment, each gecko was fitted with a harness and leash made of a lightweight, smooth dental floss, loosely tied around its waist. Caution was taken to ensure the harness was not impeding movement. Before being placed in the experimental arena, the geckos were left for a minimum of ten minutes to familiarise themselves with room temperature and the harness. The subject's movements were constrained to the centre of the experimental arena by the leash taped to the arena itself (around 15cm of range of motion) as well as through a 10cm tall cardboard visual barrier surrounding the arena. When needed, a brush was used to coax the animal back into the camera's field of view. Each animal was weighed on a precision scale (Sartorius M-Pact AX224, Sartorius AG, Goettingen, Germany, precision: 0.0001g) before and immediately after being tested. Although animals were fasted the day prior to testing, any defecation (often water) that took place between these measurements was noted.

When isothermal with the environment, the animal was placed in the test arena, described in Fig 1 for the heliothermic heating experiment and thigmothermic heating experiment (in a random order, on separate days). Immediately upon contact with the arena, a thermal video of the animal was recorded with an infrared (IR) thermal camera (FLIR T335, FOL 18 mm lens, sensitivity: < 0.05˚C; accuracy: ± 2%; IR image resolution: 320×240 pixels; IR video framerate: 10fps; FLIR Systems Inc., Wilsonville, Oregon, USA) fitted directly above the arena at a vertical distance of 30cm from its centre. This was used to monitor (during the experiment) and measure (*a posteriori*) the animal's body temperature. An additional CCTV RGB (closed circuit, colour video system MHD-CH30-130H, video framerate: 30fps) camera was set perpendicularly and at eye level to the animal, oriented towards the animal's flank, to monitor any escape attempts and to record the body posture of the animal under three different categories: elevated (just the feet in contact with the substrate), partially elevated (part of the ventral area in contact with the substrate) or flat (the animal laying the whole ventral surface flat on the substrate). The head and the tail were not considered in the posture categories.

The animal was allowed to heat up for a maximum of 10 minutes or until it reached a temperature of 35˚C at any body part (monitored through the IR camera's live feed). This is a conservative maximum threshold above the reported the thermal preference for this species (i.e. 29.8 ± 3.79˚C [55]). Hence, and since critical and voluntary thermal limits are not yet reported

for *T. mauritanica*, this threshold was defined through a pilot study where signs of discomfort (e.g. restlessness, toepads curling up) where observed when foot temperatures exceeded 35˚C.

Immediately after the heating stage, the animal was then recorded cooling down for a maximum period of 10 minutes or until the temperature across the gecko's body was homogenous and isothermal with room temperature. Only one animal was tested at each time and multiple days were given between the heliothermic and thigmothermic trials of each individual.

During each trial, the following measurements were taken every 20 seconds, starting concurrently with the thermal video recording: water temperature inside the inverted aquarium (k-type thermocouple probe), air temperature (temperature and humidity metre), body posture (via the RGB video), whether the animal moved or stayed mostly stationary within the previous 20s period, and whether the animal defecated or not (regardless of whether it was water, urates or faeces) in that same period. Additionally, maximum body temperature of the animal was monitored through the thermal camera´s live feed to prevent the animal from reaching potentially stressful temperatures (conservative threshold set for 35˚C at any body part). A trial (heating or cooling) would end after ten consecutive minutes of thermal video recording or if any part of the animal reached the maximum temperature threshold. In the latter case, the trial would be repeated if less than 3 minutes of thermal video had been attained. For this, and since such cases would only apply during the heating trials, the gecko was allowed to rest for at least 10 minutes, in order to return its body temperature to room temperature before repeating the process.

Data was always collected by the same person (FMB) with a second person (AET, GMR or VM) invigilating the test subject in the arena and managing the air and water temperatures in the room and aquaria.

## Ethics and permits

The study was performed under the regulations approved by the Committee of Animal Experimentation of the University of Porto (Portugal) under the Directive 2010/63/EU of the European Parliament. Animal sampling was carried out under the permit numbers 27497/2019/ DRNCN/DGEFF and SGIB/AF from *Instituto da Conservação da Natureza e das Florestas* (Portugal) and *Consejería de Agricultura, Ganadería y Pesca y Desarrollo Sostenible de la Junta de Andalucía* (Spain).

The described protocol was specifically devised to minimise invasiveness. All processes involving the animals were less invasive than the insertion of a needle. Furthermore, at the end of the study, all the geckos were unharmed and released back to their respective sites of capture.

## Data retrieval and statistical analyses

IR videos were analysed *a posteriori* with FLIR Tools software (version 5.13.18031.2002). Using the *Spotmeter* tool, temperature data was extracted from one frame every 20 seconds for seven different body parts: snout (tip), right eye, head (centrally, above parietal scales), mid-dorsum, right leg (knee), right foot (centrally) and tail (dorsally, above the cloaca) (Fig 2). For this analysis, skin emissivity was set at 0.96, following previous works from the authors [44], as well as a distance of the camera to the subject set to 30 cm and ambient temperature left at the software's pre-set 18˚C. All cooling videos were processed by VM and all heating videos processed by AET, since these two heat exchange processes were analysed separately. For consistency, prior to the video analysis all authors agreed on the reference points used to define the exact locations from which temperatures would be extracted in different body parts. Additionally, and to standardise the thermal video processing protocol, VM and AET analysed a

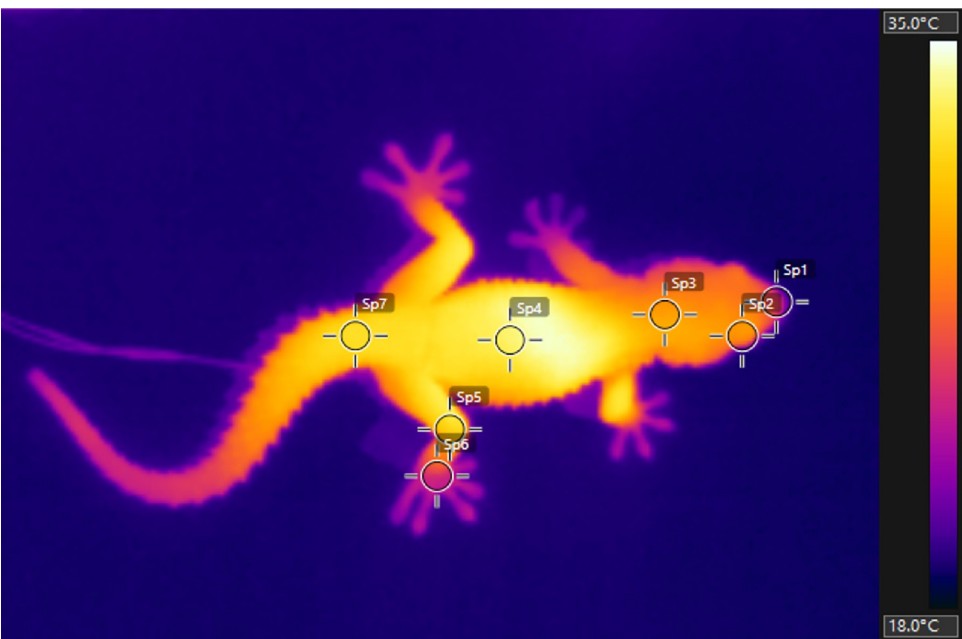

**Fig 2. Thermal image of *T. mauritanica* obtained from a frame of one of the thermal videos recorded.** Radiometric image showing the placement of the measurement points for each body part: Sp1—snout, Sp2—right eye, Sp3—head, Sp4—mid-dorsum, Sp5—right leg, Sp6—right foot and Sp7—tail.

practice sub-set of the videos together, prior to the final video analysis. A detailed description of a since enhanced version of the thermal video processing protocol is available in the online protocol repository https://dx.doi.org/10.17504/protocols.io.n2bvj358xlk5/v1 [61].

Before any statistical analysis was conducted, raw absolute temperatures were converted into cumulative temperature changes so that starting temperature change is zero (i.e. *Cumulative Temperature at time$_x$ = Temperature at time$_x$—Temperature at time$_0$*), thus correcting for potential differences in the starting temperature of the different animals and between different body parts. Heating and cooling trials were treated as separate datasets as these refer to different processes.

Paired T-tests between initial and final mass (per treatment) were carried out to explore if animals experienced significant mass losses during the trials as large changes in mass might also represent changes in the animal's thermal inertia.

Linear Mixed-Effect models were applied to each dataset independently, with cumulative temperature change as the dependent variable, using the R package *lme4* v1.1–35.1 [62]. Original full models contained all the measured variables: time, treatment (heliothermic vs thigmothermic), body part, defecation, posture, initial mass and population. Additionally, relevant interactions between variables were considered, and individual ID added as a random factor to account for the repeated measures. Time was coerced to follow a polynomial function (second-order orthogonal polynomial), since it considerably increased the fit of the model, as expected from the literature [37]. A backward stepwise regression approach [63] was used for model selection whereby nested models were compared based on a chi-squared test between the log-likelihood of the two models. This sequential process was undertaken until a minimal model was reached (independently for each heat exchange process). Once the final model was selected, residuals were verified to accomplish the assumptions of homoscedasticity and normality, and the p-values for the minimal models' explanatory variables computed with the R

package *lmerTest* v3.1.3 [64]. The effect size of each such variable was calculated with the R package *effectsize* v0.8.6 [65].

A second analysis was carried out with a data subset containing only one minute of trial, namely, the time interval 60-120s, since the highest rates of heat exchange are expected when the difference in temperature between the animal and its environment is the largest (i.e. the start of the trial). However, the first minute was discarded since the animals were often still getting used to the set-up and therefore moved considerably, resulting in a higher amount of time performing exploratory, non-thermoregulatory behaviours.

Ultimately, this allowed us to investigate the rapid (maximal) rates of temperature change in each body part by comparing the gradient of temperature change per time for each body part. The same statistical pipeline, described for the full dataset, was applied to the maximum rates dataset. An additional step included the calculation of the Ordinary Least Squares (OLS) regression slopes for the relationship between temperature change and time (i.e. the rate of temperature change) for each body part, under each treatment (heliothermy vs thigmothermy) for each heat exchange process (heating vs cooling). The slopes, intercepts and corresponding 95% confidence intervals (CI's) were calculated using the R package *lmodel2* v1.7–3 [66].

Finally, graphs were plotted using the *ggplot2* v3.4.4 package [67]. All analyses for this study were conducted in R software, version 4.3.2 [68]. All data and R scripts for statistical analyses are openly available in the following GitHub repository: https://github.com/FredericoMB/ Tarentola_Thermal_Exchange_Rates. Thermal videos available upon request to the corresponding author.

## Results

Paired T-test comparisons between initial and final mass, for each treatment, did not show any significant differences (Heliothermy: t = -0.061, p-value = 0.95, df = 78; Thigmothermy: t = -0.081, p-value = 0.93, df = 78). Hence only initial mass (i.e. henceforth "mass") was considered for the remaining analyses.

### Full dataset

Different heating and cooling patterns were observed among body parts (Fig 3). During heliothermic heating, the cumulative increment of temperature achieved a plateau around 200–300 seconds after starting the experiment (Fig 3). However, the magnitude of total temperature change differed between body parts. In general, the foot showed half the temperature increase when compared to the heating of the other body parts, while the snout showed an intermediate position. For the cooling trials, a similar but inverted pattern was observed (S2 Fig). Overall, the foot was the body part showing the least temperature decrease, while the temperature of the snout decreased 1.5 times more than that of the foot and all the remaining body parts decreased double the amount than that of the foot. In general, the heating of the heliothermic experiment tended to finish earlier than the cooling since, for heliothermic heating, individuals often achieved the 35°C threshold before reaching the 10th minute of the trial.

Contrastingly, during the thigmothermic heating experiment, the foot attained higher temperatures and much sooner than other body parts. For the foot, within the first 200 seconds, the cumulative increment of temperature was at least double than that of the second warmest body part (Fig 3). However, it is worth noting that the y-intercept of the 2nd order polynomial fitted to the foot thigmothermic heating data was considerably different from zero, thus hinting at a very rapid increase in the foot temperature within the first 20 seconds of the trial. The cooling part, however, showed the inverted pattern, as previously observed during the cooling of the heliothermic experiment. Interestingly, during heating, the mean increment of

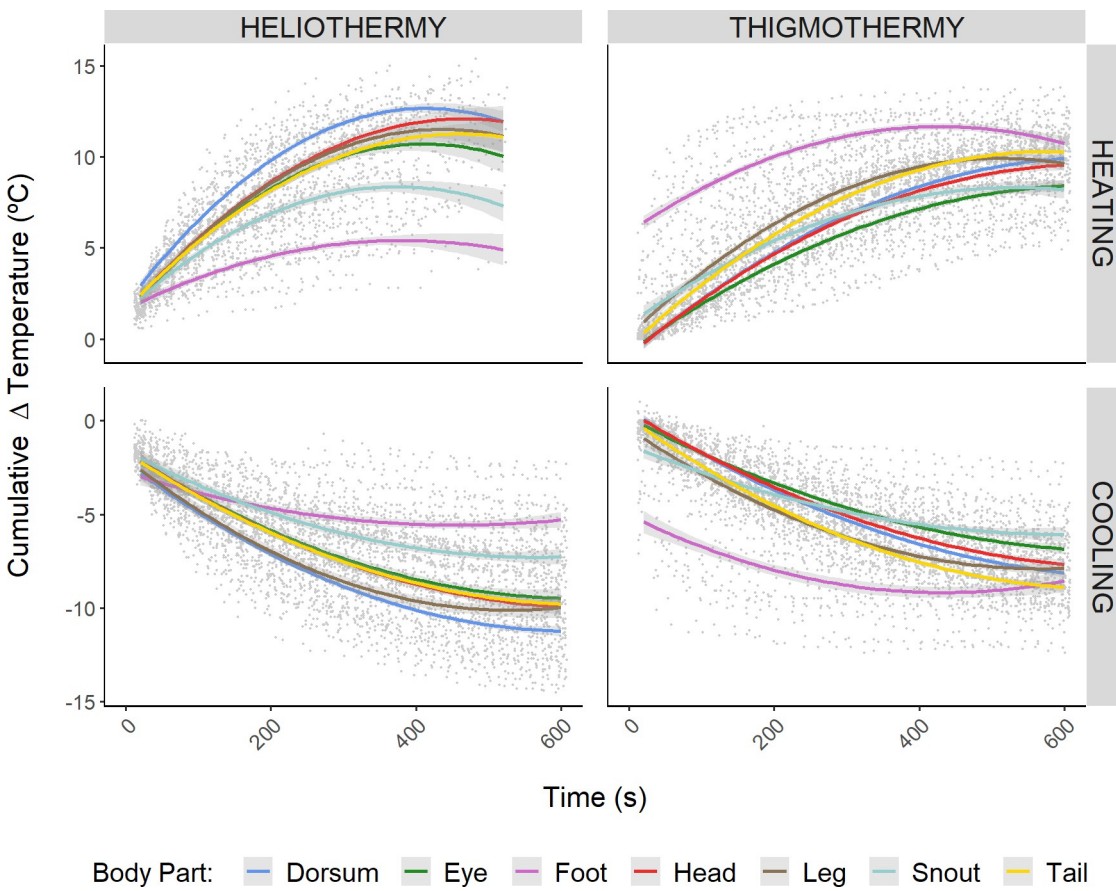

**Fig 3. Plots showing the cumulative temperature change over time.** The figure shows the cumulative temperature change (i.e. *Cumulative Temperature at time$_x$ = Temperature at time$_x$ — Temperature at time$_0$*) in both heliothermic (left column) and thigmothermic (right column) experiments, during heating (top row) and cooling (bottom row) processes. Lines represent a second order polynomial regression between the *cumulative temperature change* against *time*, for each of the measured body parts (given by the different colours) and with the shaded area around the lines representing the corresponding standard error. Points represent cumulative temperature change of each body part from each individual, shown here purely for a graphical representation of the data's dispersion.

cumulative temperature among all body parts for both heliothermic and thigmothermic experiments was very similar (10.27 ± 2.61˚C vs 9.91 ± 1.72˚C respectively). Conversely, during cooling, animals in the heliothermic experiment lost almost 1.5˚C more than during the thigmothermic experiment.

Model selection for the full dataset yielded similar results between the two heat exchange processes (heating vs cooling). Both models kept *individual ID* as a significant random factor. The heating model kept the independent variables of *time* [1$^{st}$ and 2$^{nd}$ order polynomial terms, hereafter *poly(time, 2)*], *treatment* (heliothermy vs thigmothermy), *body part*, *defecation* and *mass*, as well as the interactions between *poly(time, 2)* and *treatment*, *poly(time, 2)* and *body part*, *treatment* and *body part*, and the triple interaction between *poly(time, 2)*, *treatment* and *body part*. All such variables and interactions, with the exception of *mass*, were statistically significant (Table 1). The cooling minimal model included the same variables and interaction terms as the heating minimal model previously described, with the addition of the *posture* variable (Table 1). An ANOVA of the variables in this model, showed that again, the *mass*, but also the *poly(time, 2)* x *treatment* interaction, were not significant (Table 1).

Further exploring the magnitude of the effect size of each variable, in each minimal model, demonstrated that the *poly(time, 2)* terms had consistently large effects for both heating and cooling models (S4 Fig). Additionally, the interactions of *poly(time, 2)* with the *body part* variable, as well as with the *body part* x *treatment* variables, all showed large effect sizes (S4 Fig). Nonetheless, these variables, and especially the *time* polynomials, showed particularly broad 95% CI's (S4 Fig), displaying large variability of the effect size of these parameters.

## Maximum rates dataset

For maximum rates dataset, overall absolute cumulative temperature changes in the thigmothermic trials were both slower and less divergent among body parts, when compared to those observed for the heliothermic trials (Fig 4). Additionally, within each heating type, thermal exchange rates of different body parts were also more similar within the cooling trial than within the heating trial (Fig 5). Again, the foot showed the most divergent results among the different body parts (Fig 4) with a higher heating rate for the thigmothermic process than for the heliothermic one. Furthermore, the foot's heliothermic heating rate, as given by the absolute slope of the respective OLS regression (S3 Fig), was the lowest heating rate of any body part in any of the treatments (Fig 5). Conversely, the dorsum under heliothermic heating

**Table 1. ANOVA tables for the independent, fixed effects variables in the minimal models for each heat exchange process (heating vs cooling) in each of the datasets (full vs maximum rates).**

| Dataset | Variable(s) | HEATING | | | | COOLING | | |
|---|---|---|---|---|---|---|---|---|
| | | Deg. Freed. | F-value | Pr(>F) | | Deg. Freed. | F-value | Pr(>F) |
| **Full Dataset** | poly(Time, *2*) | 2, 6375.0 | 577.9081 | **<2.2e⁻¹⁶** | | 2, 7624.6 | 506.9119 | **<2.2e⁻¹⁶** |
| | Treatment | 1, 6387.8 | 672.9938 | **<2.2e⁻¹⁶** | | 1, 7639.5 | 1686.6309 | **<2.2e⁻¹⁶** |
| | Body Part | 6, 6373.7 | 97.3144 | **<2.2e⁻¹⁶** | | 6, 7624.7 | 186.6488 | **<2.2e⁻¹⁶** |
| | Mass | 1, 28.5 | 0.4389 | 0.5130 | | 1, 29.8 | 0.0971 | 0.7575 |
| | Defecation | 1, 3454.8 | 14.6978 | **0.0001** | | 1, 4483.3 | 76.0981 | **<2.2e⁻¹⁶** |
| | Posture | *NA* | *NA* | *NA* | | 1, 7327.9 | 28.0999 | **1.186e⁻⁷** |
| | poly(Time, 2) x Treatment | 2, 6378.3 | 87.3184 | **<2.2e⁻¹⁶** | | 2, 7625.0 | 1.5557 | 0.2111 |
| | poly(Time, 2) x Body Part | 12, 6373.3 | 67.6620 | **<2.2e⁻¹⁶** | | 12, 7624.3 | 86.9433 | **<2.2e⁻¹⁶** |
| | poly(Time, 2) x Mass | 2, 6374.7 | 4.2508 | **0.0143** | | 2, 7624.7 | 12.8476 | **2.690e⁻⁶** |
| | Treatment x Body Part | 6, 6373.7 | 633.9881 | **<2.2e⁻¹⁶** | | 6, 7624.9 | 577.4286 | **<2.2e⁻¹⁶** |
| | poly(Time, 2) x Treatment x Body Part | 12, 6373.3 | 7.7794 | **1.447e⁻¹⁴** | | 12, 7624.3 | 3.5849 | **2.324e⁻⁵** |
| **Max. Rates Dataset** | poly(Time, *1*) | 1, 759.87 | 681.482 | **<2.2e⁻¹⁶** | | 1, 630.08 | 476.3650 | **<2.2e⁻¹⁶** |
| | Treatment | 1, 761.42 | 16.837 | **4.512e⁻⁵** | | 1, 641.54 | 22.0614 | **3.232e⁻⁹** |
| | Body Part | 6, 758.16 | 14.249 | **2.002e⁻¹⁵** | | 6, 627.89 | 8.5732 | **5.662e⁻⁶** |
| | Population | *NA* | *NA* | *NA* | | 3, 15.50 | 2.8506 | 0.0714 |
| | Treatment x Body Part | 6, 757.83 | 21.014 | **<2.2e⁻¹⁶** | | 6, 627.85 | 4.6467 | **0.0001** |

**Bold** p-values indicate significance at alpha < 0.05. Note that the variable *poly(Time, 2)* in the full dataset minimal models, differs from the *poly(Time, 1)* in the minimal models for the maximum rates dataset. The former refers to two orthogonal polynomials (1st order and 2nd order) fitted, and hence uses 2 degrees of freedom (one for each curve), while the latter only fits one [1st order] polynomial only, thus only using 1 degree of freedom, as the 2nd order polynomial was deemed non-significant (Heating: $\chi^2 = 0.0508$, p = 0.8216; Cooling: $\chi^2 = 0.0159$, p = 0.8996) during the model selection process. Degrees of freedom estimated with Satterthwaite's method [69] and reported as *"Numerator d.f., Denominator d.f."*.

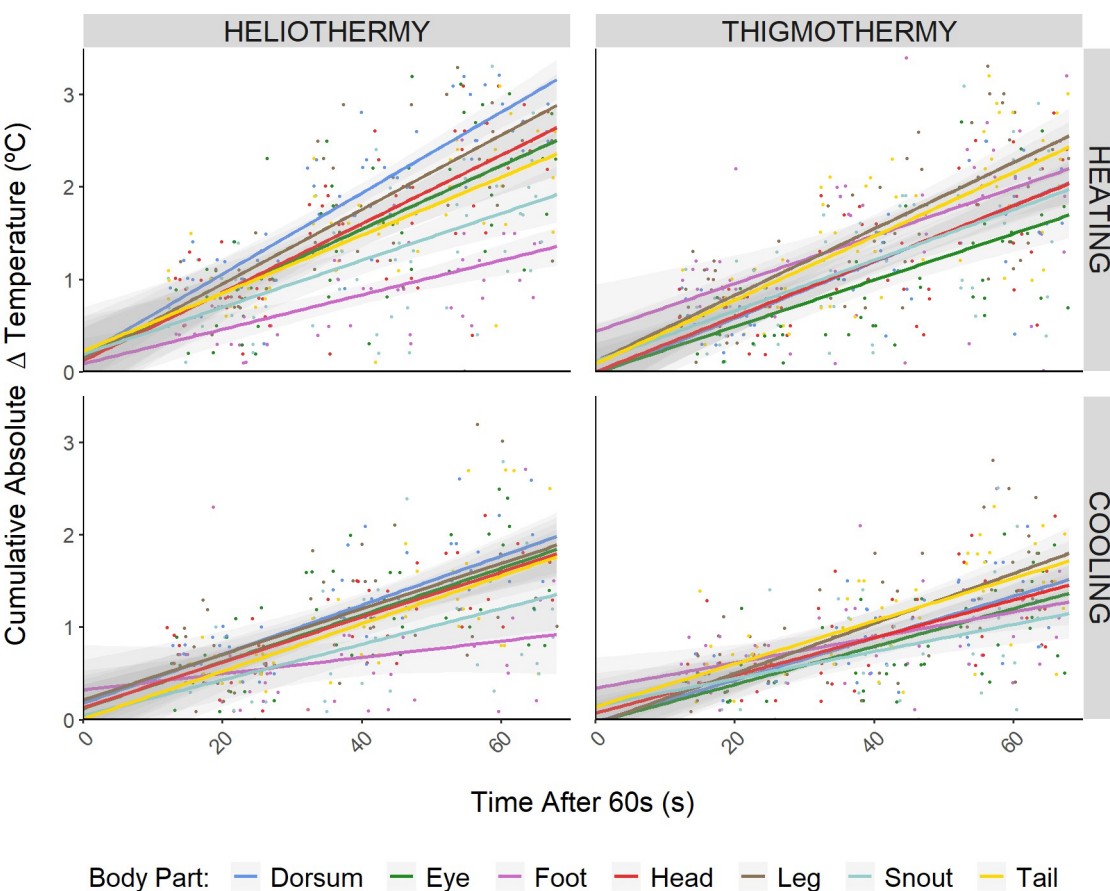

**Fig 4. Absolute cumulative temperature change per time for maximum rates dataset.** These plots show the absolute cumulative temperature change for one minute, after the first 60 seconds of the experiment (i.e. | *Temperature at time$_x$—Temperature at time$_{60}$* |; where *60s ≤ x ≤ 120s*), representing the maximum rate of heat exchange of each body part, in both heliothermic (left column) and thigmothermic (right column) experiments during heating (top row) and cooling (bottom row) processes. Lines represent the linear regression between the *cumulative absolute temperature change* against *time*, for each of the measured body parts (given by the different colours) and with the shaded area around the lines representing the corresponding standard error. Points represent the cumulative absolute temperature change for each body part of each individual.

exhibited the fastest thermal exchange rate of any body part, under any treatment (Figs 4 and S3). Additionally, the dorsum also showed the most marked difference between the heating rates of the two treatments (Fig 5).

The OLS regression showed no intercept deviating significantly from zero (Table 2), but slope values were more variable. The strength of the linear relationship varied extensively ($0.06 ≤ R^2$ of OLS regressions $≤ 0.74$) depending on the body part, treatment and heat exchange process (Table 2) and was generally weak (overall mean $R^2 ± SE = 0.44 ± 0.03$). Heating showed stronger relationships than cooling (mean $R^2 ± SE$ for heating $= 0.48 ± 0.04$ vs for cooling $= 0.41 ± 0.04$). The foot consistently had a very weak fit, showing the lowest $R^2$ values (mean $R^2 ± SE = 0.18 ± 0.04$) of all other body parts (Table 2), as opposed to the dorsum, which showed the best fit in all regressions (Table 2), with its lowest fit for thigmothermic cooling ($R^2 = 0.59$) and its highest fit for heliothermic heating ($R^2 = 0.74$) demonstrating a good overall fit on all its models (mean $R^2 ± SE = 0.65 ± 0.03$).

Model selection on the linear mixed effects models for the heating and cooling maximum rates datasets produced simpler minimal models than those attained for the full dataset (Table 1). Again, both minimal models kept *individual ID* as a significant random factor, and

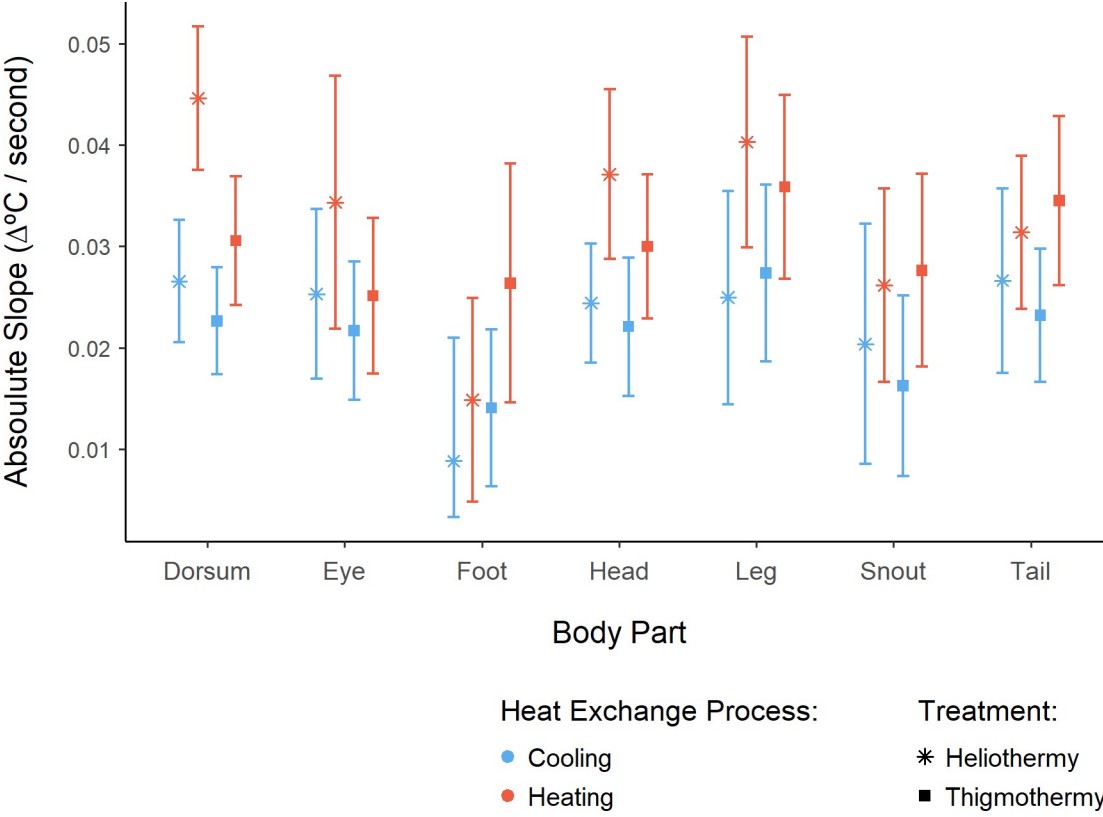

**Fig 5. Slopes and corresponding 95% confidence intervals, for the maximum heating (red) and cooling (blue) rates, under the heliothermic (star) or thigmothermic (square) treatments.** These slopes were obtained from the Ordinary Least Square (OLS) regression models. Absolute values were used to facilitate direct comparison of the slope's magnitude, since the cooling and heating regressions would have opposite trends given the directly contrary directionality of temperature change over time for these two processes.

both models differed only by one variable—in this case, *population*, which was absent from the maximum heating rates model but present, albeit ultimately non-significant, in the maximum cooling rates minimal model (Table 1). Much like the full dataset minimal models described previously, *time* proved to be a highly significant variable (*poly(time, 1)*) for the maximum rates models as well. However, in this case, the $2^{nd}$ order polynomial was deemed non-significant during the model selection process (Heating: $\chi^2 = 0.0508$, p = 0.8216; Cooling: $\chi^2 = 0.0159$, p = 0.8996). The remaining significant variables, present on both heating and cooling maximum rates minimal models, were *treatment*, *body part* and their interaction (Table 1).

Again *time* showed the largest effect size by an entire order of magnitude, while all other variables had much smaller and similar effect sizes (S5 Fig). Furthermore, in addition to showing the largest effect size, *time* once more showed the largest range in the magnitude of its effect.

## Discussion

Despite the generally acknowledged vital role that thermal exchange rates play in ectotherm ecophysiology [13,14], traditional methods for estimating these often fail to provide realistic insights into the integrative interplay between the environmental, behavioural and physiological variables which modulate them. This was likely due to the technological limitations when most such classic thermal exchange rate studies were developed, particularly when applied to

**Table 2. Model output table of the Ordinary Least Squares (OLS) regressions for the maximum thermal exchange rates dataset.** The slope of the OLS regressions represents the rate of heat transfer.

| Treatment | Body Part | HEATING | | | | | COOLING | | | | |
|---|---|---|---|---|---|---|---|---|---|---|---|
| | | Slope | Slope 95% CIs | Intercept | Intercept 95% CIs | $R^2$ | Slope | Slope 95% CIs | Intercept | Intercept 95% CIs | $R^2$ |
| Heliothermy Experiment | Eye | 0.034 | 0.022 0.047 | 0.159 | -0.379 0.697 | 0.35 | -0.025 | -0.034 -0.017 | -0.119 | -0.472 0.234 | 0.46 |
| | Snout | 0.026 | 0.017 0.036 | 0.147 | -0.265 0.560 | 0.35 | -0.020 | -0.032 -0.009 | 0.020 | -0.482 0.521 | 0.25 |
| | Head | 0.037 | 0.029 0.045 | 0.118 | -0.243 0.478 | 0.59 | -0.024 | -0.030 -0.019 | -0.131 | -0.376 0.114 | 0.61 |
| | Dorsum | 0.045 | 0.038 0.052 | 0.138 | -0.167 0.444 | 0.74 | -0.027 | -0.033 -0.021 | -0.176 | -0.428 0.075 | 0.62 |
| | Leg | 0.040 | 0.030 0.051 | 0.144 | -0.305 0.593 | 0.55 | -0.025 | -0.035 -0.014 | -0.195 | -0.630 0.240 | 0.37 |
| | Foot | 0.015 | 0.005 0.025 | 0.135 | -0.299 0.569 | 0.15 | -0.009 | -0.021 0.003 | -0.320 | -0.805 0.166 | 0.06 |
| | Tail | 0.031 | 0.024 0.039 | 0.219 | -0.107 0.545 | 0.57 | -0.027 | -0.036 -0.018 | 0.038 | -0.345 0.422 | 0.43 |
| Thigmothermy Experiment | Eye | 0.025 | 0.017 0.033 | -0.014 | -0.346 0.318 | 0.44 | -0.022 | -0.028 -0.015 | 0.091 | -0.204 0.386 | 0.44 |
| | Snout | 0.028 | 0.018 0.037 | 0.098 | -0.313 0.510 | 0.37 | -0.016 | -0.025 -0.007 | -0.052 | -0.438 0.334 | 0.23 |
| | Head | 0.030 | 0.023 0.037 | -0.004 | -0.312 0.305 | 0.57 | -0.022 | -0.029 -0.015 | 0.021 | -0.274 0.316 | 0.45 |
| | Dorsum | 0.031 | 0.024 0.037 | -0.034 | -0.308 0.240 | 0.65 | -0.023 | -0.028 -0.017 | 0.024 | -0.204 0.253 | 0.59 |
| | Leg | 0.036 | 0.027 0.045 | 0.111 | -0.281 0.503 | 0.52 | -0.027 | -0.036 -0.019 | 0.058 | -0.319 0.435 | 0.45 |
| | Foot | 0.026 | 0.015 0.038 | 0.409 | -0.102 0.919 | 0.26 | -0.014 | -0.022 -0.006 | -0.323 | -0.658 0.013 | 0.23 |
| | Tail | 0.035 | 0.026 0.043 | 0.086 | -0.276 0.448 | 0.54 | -0.023 | -0.030 -0.017 | -0.138 | -0.424 0.147 | 0.49 |

small ectotherms. However, with the onset of new tools such as thermal imaging [53], such evidence may be revised in the light of fine-tuned thermal data, at both spatial and temporal scales, in order to provide a more in-depth and integrative perspective into these and other thermophysiological phenomena [70].

Here a replicable methodology to study thermal physiology in small to medium sized ectotherms is provided. This protocol allows examining thigmothermy and heliothermy separately, while under the same (i.e. comparable) methodological framework. Hence, this will enable disentangling the effect of thermal exchange strategies and of their underlying ecophysiological and behavioural variables. Ultimately, it will allow presenting a finer view into the thermal ecology of the studied animals.

Another major contribution is the ability to quantify thigmothermic heat exchange, a thermal strategy mostly neglected in the literature due to the difficulties in incorporating it into classical studies (but see [2]). This study opens an avenue to examine hundreds of species with thigmothermic behaviours, as well as to test if species previously thought to be strict heliotherms could also complement their preferred thermoregulatory strategy with thigmothermic thermal exchanges.

Specifically for the model species, our results reveal different thermal patterns depending on the body part (Figs 3 and 4), providing further evidence for regional heterothermy [71] in

small lizards, in line with what has been reported in other squamates (e.g. [41,42]). Of the body parts studied, the foot experienced the slowest and smallest cumulative change in temperature in the heliothermic heating and cooling processes (Fig 3). This was expected since, under heliothermy, the foot acts as a location of heat loss to prevent overheating. Being in direct contact with the cold substrate and given its small size, it is likely that the rate at which it is receiving heat from the rest of the body may be very similar to the rate at which it is losing heat to the cold substrate, thus buffering the range of temperatures experienced. A similar pattern is also observed for the thigmothermic cooling of the foot. However, having the slowest rate of the body parts monitores, an intercept considerably different from zero, and a nearly straight-line pattern (i.e. the least curved of all the body parts' regression lines), hints at its high sensitivity to temperature change. Additionally, the foot was the body part showing the fastest heating rate during the thigmothermic heating trials, as expected due to its small size and concordant with previous comments on the sensitivity to temperature. Its small size, and hence small inertia, suggest that it is less likely to accumulate "thermal history" (i.e. be less affected by past states and conditions) than other body parts and quickly equilibrate with its environment [48,72,73].

The differences between the foot's cooling processes (post-heliothermic vs post-thigmothermic heating) are also worth noting. They could be purely biophysical—i.e. the foot had different thermal starting points simply because the resulting pattern of regional heterothermy differed between the two heating treatments—but may result from the gecko's ability to identify the main heat source and adjust its blood circulation accordingly in order to optimise the heat distribution along the body [74,75]. If this were true, it would mean that the animals' physiology may be more "focused" on maximising heat gain rather than on minimising heat loss, as often reported in other reptiles [76] (reviewed by [15]). Interestingly though, the leg follows the pattern observed in the remaining body parts. This may suggest vasoconstriction in the wrists as well as the presence of specific vascularisation patterns of the limbs (as already reported for other geckos [77]). The latter may be linked to the maintenance of proper blood irrigation and hence of the adhesive function of the lamellae in geckos [78]. Alternatively, it may function as counter-current heat exchange systems, as found in other animals [31,79,80], allowing for a fine control of thermal exchange at this site.

Regarding the thermal exchange rates at other body parts, during the heliothermic heating experiment, the dorsum showed the fastest heating rate as well as the widest thermal range (Figs 3 and 5). This was expected as reptiles utilise their dorsum as a main area for radiant heat absorption. Hence, they may capitalise on the dorsum's large thermal inertia to utilise it as a thermal reservoir from which heat can be redistributed to other parts of the body. Furthermore, the dorsum was the body part that experienced the fastest and largest temperature shift during the post-heliothermic cooling trial (S2 Fig), further supporting the thermal reservoir hypothesis since it can accumulate excess heat when basking, and then redistribute it once the external heat source was no longer available.

As for the thigmothermic heating and cooling, the dorsum followed an intermediate/ average rate and range of temperature, which is in line with its large inertia and the fact that it is not directly an area of heat acquisition. The eye—shown to be a good proxy of internal body temperature for this [44] and other [43] lizards—and the snout also show comparatively lower thermal ranges and slower heat exchange rates than other body parts (except the foot). This is compatible with the presence of surface-to-core temperature gradients, but might also result from evaporative heat loss [31], which would allow for faster cooling of these areas, even while the whole animal is heating up. The latter has been hypothesised as a strategy to avoid the overheating of sensitive, vital organs such as the brain [31,42,81]. If, however, evaporation from ocular and nasal surfaces led to significantly increased rates of heat loss in these structures,

then this should also be reflected in the cooling trials as faster cooling rates of these body parts. In fact, the opposite was observed. This further reinforces the idea that inner body temperature tends to be more stable and takes longer to respond to changes in the external thermal environment, even in a smaller-bodied animal, but could also result from minimal ocular evaporation due to the presence of a scale (the brille) covering the eyes–a common feature among geckos [82].

In general, for all body parts but the foot, heliothermic processes (both heating and cooling) were faster and showed wider thermal ranges (Figs 3 and 5). Nonetheless, this might be artefactual since the two set-ups were not emitting the same power. Hence, the biological significance of this difference cannot be discussed here. Arguably, though, this may still represent an ecologically realistic scenario since, if a surface is "too hot for comfort", the animal will likely not remain still as it may burn its feet [83]. Contrastingly, even in very hot sun, a cold ectotherm could be found basking for the amount of time needed to obtain a temperature within its set-point range [36,84], as long as the substrate itself is not too hot. Nonetheless, future deployments of this experimental framework may consider adjusting the power and/or distance of the heat source so that the resultant operative temperature is the same between treatments. Furthermore, colouration should also be monitored, as it can influence the rate of radiant heat transfer [85] (but see [45,46]). This may be of particular relevance for species that exhibit rapid physiological colour change [86] whereby the animal can rapidly darken or lighten its skin according to its ecophysiological needs, as reported even for the species tested here [87,88].

## Conclusion

Ultimately, with this study we present a new and replicable methodological procedure to infer heat exchange through heliothermy and thigmothermy in ectotherms. This novel method successfully integrated environmental, physiological and behavioural variables, within an experimental context, in order to precisely describe thermal exchange rates at a high spatial (across different body parts) and temporal resolution while relying on new technologies that minimise the invasiveness during data collection. Moreover, we provide a proof-of-concept of this methodology by testing it on a crepuscular and nocturnal ectotherm, where we were able to unravel the complexity and importance of regional heterothermy, showing that body parts within the body of a small lizard can behave differently depending on the heat source available, even in a short timeframe.

The ease of replicability of the described set-up, along with its flexibility, could enable future studies to adapt this method to investigate other factors (e.g. colouration, wind, body shape and condition, etc) potentially affecting thermal exchange rates and body heat distribution in ectotherms. The design may easily be extended to other organismal models, such as amphibians and non-flying arthropods, as well as simulate different environmental contexts (e.g. climate change, wind, presence of conspecifics, predation risk, etc). More importantly, this methodology is expected to further the knowledge of terrestrial ectotherm thermal physiology and test alternative hypotheses, representing a valuable tool for addressing thermal ecology questions—classical and new—with innovative, integrative and less stressful approaches.

## Supporting information

**S1 Fig. Mass (at start vs at the end of each treatment trial) and overall Snout-to-Vent Length (SVL) of all the tested animals.** The line in the middle of the boxplot represents the median value, while the "X" represents the mean value. No significant differences between

initial and final mass were observed nor were there any differences in mass between treatments.
(TIF)

**S2 Fig. Thermal exchange rate graphs per body part.** Heating (left column) and cooling (right column) profiles for the different body parts, under heliothermic (full line, full points) and thigmothermic (dashed line, empty points) treatments. 2nd order polynomial fitted to demonstrate the trend in the data.
(TIF)

**S3 Fig. Graphical representation of the OLS regression lines (and 95% CI's in grey shade) for the maximum rates of cooling (blue) and heating (red).** Absolute rates are plotted in order to facilitate the comparison between the magnitude of the slope between the heating and cooling processes.
(TIF)

**S4 Fig. Effect size plot for the variables of the minimal linear mixed effects model for the full heating and cooling rates datasets.**
(TIF)

**S5 Fig. Effect size plot for the variables of the minimal linear mixed effects model for the heating and cooling maximum rates datasets.**
(TIF)

## Acknowledgments

We thank *BIOPOLIS-CIBIO* research centre for providing the facilities and support for the experiments as well as members and collaborators of the *Functional Biodiversity* research group for assistance in the field and useful feedback on initial drafts of the manuscript.

## Author Contributions

**Conceptualization:** Gabriel Mochales-Riaño, Frederico M. Barroso, Valéria Marques, Alexandra E. Telea, Miguel A. Carretero.

**Data curation:** Gabriel Mochales-Riaño, Frederico M. Barroso.

**Formal analysis:** Gabriel Mochales-Riaño, Frederico M. Barroso, Marco Sannolo.

**Funding acquisition:** Catarina Rato, Miguel A. Carretero.

**Investigation:** Gabriel Mochales-Riaño, Frederico M. Barroso, Valéria Marques, Alexandra E. Telea.

**Methodology:** Gabriel Mochales-Riaño, Frederico M. Barroso, Valéria Marques, Alexandra E. Telea.

**Resources:** Catarina Rato.

**Supervision:** Marco Sannolo, Catarina Rato, Miguel A. Carretero.

**Visualization:** Gabriel Mochales-Riaño, Frederico M. Barroso.

**Writing – original draft:** Gabriel Mochales-Riaño, Frederico M. Barroso.

**Writing – review & editing:** Gabriel Mochales-Riaño, Frederico M. Barroso, Valéria Marques, Alexandra E. Telea, Marco Sannolo, Catarina Rato, Miguel A. Carretero.

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
