## [Decision Letter · Decision Letter 0]

11 Nov 2024

PONE-D-24-45585Novel method to investigate thermal exchange rates in small, terrestrial ectotherms: A proof-of-concept on the gecko *Tarentola mauritanica*PLOS ONE

Dear Dr. Barroso,

Thank you for submitting your manuscript to PLOS ONE. After careful consideration, we feel that it has merit but does not fully meet PLOS ONE’s publication criteria as it currently stands. Therefore, we invite you to submit a revised version of the manuscript that addresses the points raised during the review process.

We have received two reviews, however, one of them is not particularly useful.

Firstly, while the method is novel, it would be strengthened by comparing it with similar thermography studies to better define its unique contributions. Additionally, the complex setup for isolating heliothermy and thigmothermy may need further justification, as a simpler approach could achieve comparable outcomes. The focus on T. mauritanica males limits broader applicability; addressing how this method might extend to other species and contexts would enhance its ecological relevance.

The statistical analysis is robust but complex, and summarizing the main effects with biological relevance would improve clarity. Additionally, observed regional heterothermy findings could be better contextualized within natural thermoregulation strategies. Finally, more details on managing potential stress from harnesses would reinforce the ethical considerations of this study.

Overall, these revisions would strengthen your work and enhance its contribution to thermoregulatory studies in ectotherms.

We look forward to receiving your revised manuscript.

Kind regards,

Edvard Mizsei

Academic Editor

PLOS ONE

Journal Requirements: When submitting your revision, we need you to address these additional requirements. 1. Please ensure that your manuscript meets PLOS ONE's style requirements, including those for file naming. The PLOS ONE style templates can be found at https://journals.plos.org/plosone/s/file?id=wjVg/PLOSOne_formatting_sample_main_body.pdf and https://journals.plos.org/plosone/s/file?id=ba62/PLOSOne_formatting_sample_title_authors_affiliations.pdf 2. Please review your reference list to ensure that it is complete and correct. If you have cited papers that have been retracted, please include the rationale for doing so in the manuscript text, or remove these references and replace them with relevant current references. Any changes to the reference list should be mentioned in the rebuttal letter that accompanies your revised manuscript. If you need to cite a retracted article, indicate the article’s retracted status in the References list and also include a citation and full reference for the retraction notice.

Reviewers' comments:

Reviewer's Responses to Questions

**Comments to the Author**

1. Is the manuscript technically sound, and do the data support the conclusions?

Reviewer #1: Yes

Reviewer #2: Yes

2. Has the statistical analysis been performed appropriately and rigorously? 

Reviewer #1: Yes

Reviewer #2: Yes

3. Have the authors made all data underlying the findings in their manuscript fully available?

Reviewer #1: Yes

Reviewer #2: Yes

4. Is the manuscript presented in an intelligible fashion and written in standard English?

Reviewer #1: Yes

Reviewer #2: Yes

5. Review Comments to the Author

Reviewer #1: This article is purely methodological. Rather than testing any particular hypothesis, it explores a novel technique to study heat acquisition and loss via thigmothermy and radiation in small ectotherms, which overcomes some limitations of previous setups. The authors use large tanks filled with water to buffer temperature, resorting to water pumps to prevent thermal stratification of the water column. Also, the animal is allowed to move to a certain extent, as it is leashed to a harness rather than immobilized as in most techniques used so far. The authors used similar (although not identical due to an alleged lack of proper material, which is a flaw, although not necessarily invalidating the technique) setups to test thigmothermy (with either hot or cold water as a source or sink of heat) and heliothermy (with water used as a buffer and a bulb as a source of heat). Also, a thermal camera was used, and temperature was recorded from a series of standardized body parts in a number of individuals, so as to ascertain how different body areas are involved in heating and cooling processes in each case. The animal was also monitored for behavioral activity to test any potential effect of movement.

The technique seems to serve the purpose it is intended for, and has indeed potential to further the study of thermoregulation and heat exchange processes in ectotherms to a fine scale. Also, it is arguably suitable to test hypotheses so far complicated to investigate, such as those involving the role of activity and behavior on thermal ecology of animals. That being said, the setup described is bulky, replicable but not necessarily easy to optimize, but that will of course depend on the logistic limitations of each researcher interested in using it, and is by no means a criticism to its functionality. I personally believe this methodology can be feasible and useful.

For the most part, the methodology is accurately described, although some minor points could be better explained. The text is in general well written, but the English needs some tweaking here and there. To my mind, the introduction and the discussion are perhaps a bit too long, especially for an article which lacks any hypothesis and is purely methodological. Also, some concepts are used in a way that is not as accurate as expected. I have highlighted below some specific points where the manuscript can be somewhat improved.

Line 24: These adaptations do not necessarily “optimize” heat exchange, as they might be compromised with others and/or simply fail to do so. I recommend a milder wording.

Line 28: Technically speaking, “heliothermy” involves heat gain from solar radiation, not just any heat source.

Line 45: “such as most/many/certain reptiles”, as many others are thermoconformers.

Line 49: I am not sure reptiles actively “emphasise” that. Maybe “reptiles depend on behavioural thermoregulation…” or something along those lines would be a more accurate wording.

Line 69: Maybe “function” or “be involved/engaged” instead of the active voice “engage”?

Line 71: Again, the adjective “undesirable” conveys too strong a human emotion. I would suggest a different wording, more objective, such as “counter-productive” or “an inherent loss of body temperature which often hampers the attainment of the thermal optimum”.

Line 75: Please. see my former comment on the word “undesirable”.

Lines 89-90: “becomes a radiant heat source itself”.

Line 90: Delete the comma after “hence”.

Line 91: “via/through/by means of radiation”.

Line 93: “of the temperature differential”.

Lines 94-95: “can actively select a substrate either warmer or cooler than itself”.

Line 97: Delete “one is”.

Line 98: “see works by Ortega et al.”.

Line 134: A comma after “Here” would be recommendable.

Line 147: “common wall”, with lowercase initials.

Lines 155-157: Why those geographically disparate locations? That does not seem necessary considering the aims of the study. Were these urban populations? Please, clarify what the numbers between parentheses are. I would say Évora coordinates are 35.352, -8.018. Does the journal accept coordinates in units other than degrees, minutes and seconds?

Line 162: Maybe “consisting of” rather than “composed of”?

Lines 164-165: A brief description of how the animals were sexed could be appropriate.

Line 166: Were these differences in body sizes coincidental, or was the selection not random, but focused on animals of different sizes? Given the aims of the study, I do not believe this is fundamental, but it should be clarified.

Line 180: “the heliothermy experiment”. I would not say the experiment is heliothermic. Here and throughout the manuscript, similar utterances should be reworded accordingly.

Line 181: “inside which a smaller (34Lx23Wx20H cm) aquarium was placed”.

Line 186: “the thickness of the lass was constant”.

Line 196: In the rest of the text, you spell this word as “utilise”, not “utilize”. Both are correct, but correspond to different standards. For consistency’s sake, please use the same standard all along.

Line 198: “they are in cooling mode”.

Line 200: The use of semicolons is highly restrictive in English. A period is preferable here.

Line 203: “the thigmothermy experiment”.

Lines 222-223: What harness and leash do you mean? What video recordings? The reader knows nothing about this at this point, as it is only explained after line 230.

Line 231: “the experimental arena”.

Line 232: “familiarize themselves with room temperature and the harness”.

Line 236: “on a precision scale”.

Line 238: Delete “for”.

Line 268: Technically speaking, feces are not excretion.

Line 292: “from one frame every 20 seconds”.

Line 424: “among body parts”.

Line 426: “were also more similar…”. The rest of the section is written using past tenses, so for consistency’s sake that should be maintained.

Line 427: Remove the comma after “results”.

Line 428: Add a comma before “with a higher…”.

Line 484: “fail to provide”.

Line 496: “it will allow”.

Line 499: “difficulties in incorporating”.

Line 511: Delete the comma after “body”.

Line 520: What is “thermal history”? It should be briefly explained.

Line 526: “If this were true”.

Line 561: Add a comma after “Nonetheless”.

Line 569: “operative temperature” is a specific concept that refers to the temperature a non-thermoregulating animal would achieve in a given situation, usually resorting to unanimated physical animal models. Is that what you mean here?

Line 571: The closing parenthesis should be added.

Line 582: “testing it on”. This gecko is crepuscular and nocturnal.

Reviewer #2: As the authors indicate, this study introduces a pioneering experimental technique that employs thermography as a less invasive alternative for precisely measuring body temperatures in thermoregulating ectotherms when quantifying thermal exchange rates. This methodology enhances the integration of behaviour and physiology, providing higher temporal and spatial resolution in measuring body temperatures of thermoregulating ectotherms. The study presents findings from original, well-designed research. The experiments and statistical analyses are conducted to a high technical standard and described thoroughly. Therefore, I recommend the article for publication in its current form.

6. PLOS authors have the option to publish the peer review history of their article (what does this mean?). If published, this will include your full peer review and any attached files.

Reviewer #1: No

Reviewer #2: No

---

## [Author Response · Author response to Decision Letter 0]

6 Dec 2024

As outline in the Response letter to Reviewers, we hereby also present, below each of the Editor’s and Reviewers’ comments, the revisions made to address the review.

Editor’s Comments:

Firstly, while the method is novel, it would be strengthened by comparing it with similar thermography studies to better define its unique contributions. 

- We have made our best attempt at integrating studies resorting to this methodology for ectotherm thermal ecology. However, it must be stressed that references are scarce and mostly focused on using radiometric photos (not video) to infer body temperatures at specific time points, and not to calculate thermal exchange rates.

Additionally, the complex setup for isolating heliothermy and thigmothermy may need further justification, as a simpler approach could achieve comparable outcomes. 

- We acknowledge that the description of the setup is complex, but to extract reliable information of thermoregulation using one heat source while excluding the other requires this kind of design, unless the editor has more specific suggestions, which we would gladly consider for future deployments of this methodology. We have, nevertheless, made an extra effort to simplify the descriptions. 

The focus on T. mauritanica males limits broader applicability; addressing how this method might extend to other species and contexts would enhance its ecological relevance.

- We resorted to using only males in order to facilitate comparisons without the confounding effect of reproductive physiology between the sexes and within females (gravid vs non-gravid). Nevertheless, the exact same set-up could be applied to females of this species, or any other similarly sized lizard, as briefly mentioned in the manuscript. In fact, other ongoing studies by the Authors are using this set-up to explore the effect of gravidity on the thermal exchange rates and patterns of regional heterothermy in a lacertid lizard. 

The statistical analysis is robust but complex, and summarizing the main effects with biological relevance would improve clarity. Additionally, observed regional heterothermy findings could be better contextualized within natural thermoregulation strategies. 

- As for the lab design, the statistics are intrinsically complex, but we hope to have improved the clarity of the description in this second version.

Finally, more details on managing potential stress from harnesses would reinforce the ethical considerations of this study.

-We have clarified this further in the manuscript.

Reviewer 1’s Comments:

This article is purely methodological. Rather than testing any particular hypothesis, it explores a novel technique to study heat acquisition and loss via thigmothermy and radiation in small ectotherms, which overcomes some limitations of previous setups. The authors use large tanks filled with water to buffer temperature, resorting to water pumps to prevent thermal stratification of the water column. Also, the animal is allowed to move to a certain extent, as it is leashed to a harness rather than immobilized as in most techniques used so far. The authors used similar (although not identical due to an alleged lack of proper material, which is a flaw, although not necessarily invalidating the technique) setups to test thigmothermy (with either hot or cold water as a source or sink of heat) and heliothermy (with water used as a buffer and a bulb as a source of heat). Also, a thermal camera was used, and temperature was recorded from a series of standardized body parts in a number of individuals, so as to ascertain how different body areas are involved in heating and cooling processes in each case. The animal was also monitored for behavioral activity to test any potential effect of movement.

The technique seems to serve the purpose it is intended for, and has indeed potential to further the study of thermoregulation and heat exchange processes in ectotherms to a fine scale. Also, it is arguably suitable to test hypotheses so far complicated to investigate, such as those involving the role of activity and behavior on thermal ecology of animals. That being said, the setup described is bulky, replicable but not necessarily easy to optimize, but that will of course depend on the logistic limitations of each researcher interested in using it, and is by no means a criticism to its functionality. I personally believe this methodology can be feasible and useful.

For the most part, the methodology is accurately described, although some minor points could be better explained. The text is in general well written, but the English needs some tweaking here and there. To my mind, the introduction and the discussion are perhaps a bit too long, especially for an article which lacks any hypothesis and is purely methodological. Also, some concepts are used in a way that is not as accurate as expected. I have highlighted below some specific points where the manuscript can be somewhat improved.

- Firstly, we thank Reviewer 1 for his/her feedback. In addition to integrating the specific suggestions made by Reviewer 1 (see below), we have gone through the manuscript to adjust the English (namely to standardise UK/US spelling inconsistencies). While also considering the requests of the editor to further clarify some aspects, we have nonetheless made an effort to further streamline the introduction and discussion of the manuscript.

Line 24: These adaptations do not necessarily “optimize” heat exchange, as they might be compromised with others and/or simply fail to do so. I recommend a milder wording.

- Changed “optimize” to “modulate”

Line 28: Technically speaking, “heliothermy” involves heat gain from solar radiation, not just any heat source.

- Specified it as radiant heat gained from the sun

Line 45: “such as most/many/certain reptiles”, as many others are thermoconformers.

- Added “many”

Line 49: I am not sure reptiles actively “emphasise” that. Maybe “reptiles depend on behavioural thermoregulation…” or something along those lines would be a more accurate wording.

- Changed to “highlight”

Line 69: Maybe “function” or “be involved/engaged” instead of the active voice “engage”?

- Changed to “be engaged”

Line 71: Again, the adjective “undesirable” conveys too strong a human emotion. I would suggest a different wording, more objective, such as “counter-productive” or “an inherent loss of body temperature which often hampers the attainment of the thermal optimum”.

- Changed to “counter-productive”

Line 75: Please. see my former comment on the word “undesirable”.

- Changed to “disadvantageous”

Lines 89-90: “becomes a radiant heat source itself”.

- Changed accordingly

Line 90: Delete the comma after “hence”.

- Done

Line 91: “via/through/by means of radiation”.

- Changed to “by means of radiation”

Line 93: “of the temperature differential”.

- Changed accordingly

Lines 94-95: “can actively select a substrate either warmer or cooler than itself”.

- Changed accordingly

Line 97: Delete “one is”.

- Done

Line 98: “see works by Ortega et al.”.

- Done

Line 134: A comma after “Here” would be recommendable.

- Added

Line 147: “common wall”, with lowercase initials.

- Changed accordingly

Lines 155-157: Why those geographically disparate locations? That does not seem necessary considering the aims of the study. Were these urban populations? Please, clarify what the numbers between parentheses are. I would say Évora coordinates are 35.352, -8.018. Does the journal accept coordinates in units other than degrees, minutes and seconds?

- The disparity in geographical locations stems from the fact that animals collected for this study were sampled as part of a broader project on Tarentola mauritanica ecology, for which a more comprehensive sampling was required. Nevertheless, these different populations are thermally distinct, meaning animals could display differences in their thermoregulatory behaviour and/or physiology. Furthermore, by using animals from different populations, we were able to test or methods in animals from a broader size range. Ultimately, our objective was to test our method on sample as diverse as possible hence, adding the population variable aided in increasing such diversity.

Indeed Évora’s coordinates were missing a “-“. It’s now rectified, and we now clarified in the text that the numbers in parentheses refer to coordinates.

As for the journal’s coordinate formatting/units, we were unable to find any specific information on the submission guidelines.

Line 162: Maybe “consisting of” rather than “composed of”?

- Changed accordingly

Lines 164-165: A brief description of how the animals were sexed could be appropriate.

- Clarified in the text.

Line 166: Were these differences in body sizes coincidental, or was the selection not random, but focused on animals of different sizes? Given the aims of the study, I do not believe this is fundamental, but it should be clarified.

- Selection was random within population, however we sampled from multiple populations (chosen haphazardly to maximise coverage of the range of the species in the Iberian Peninsula) because, in this species, geckos may attain quite different sizes. Hence, by using animals from several populations we were then able to obtain an overall size range that is both broad and representative of this species. Still, within population, there is a degree of body size variability, which contributed to the diversity sizes in the sample.

We have now clarified this in the text.

Line 180: “the heliothermy experiment”. I would not say the experiment is heliothermic. Here and throughout the manuscript, similar utterances should be reworded accordingly.

- We fully agree that since the animals are not heating via the sun, the experiment is not truly heliothermic. However, the devised set-up was precisely designed to simulate the radiant heat gain that occurs during solar exposure (i.e. a heliothermic scenario). We have now clarified in the text that the set-up is a simulation of heliothermy and that further instances of the use of such term, with regards to the experiment, refer to this simulation scenario and not a true heliothermic where the animal is heating under the sun. 

Line 181: “inside which a smaller (34Lx23Wx20H cm) aquarium was placed”.

- Done.

Line 186: “the thickness of the lass was constant”.

- Changed.

Line 196: In the rest of the text, you spell this word as “utilise”, not “utilize”. Both are correct, but correspond to different standards. For consistency’s sake, please use the same standard all along.

- Changed accordingly, and doublechecked throughout the text to ensure consistency.

Line 198: “they are in cooling mode”.

- Changed.

Line 200: The use of semicolons is highly restrictive in English. A period is preferable here.

- Changed.

Line 203: “the thigmothermy experiment”.

- Changed.

Lines 222-223: What harness and leash do you mean? What video recordings? The reader knows nothing about this at this point, as it is only explained after line 230.

- Generalised and clarified that it is explained in more detail further ahead in the text.

Line 231: “the experimental arena”.

- Changed.

Line 232: “familiarize themselves with room temperature and the harness”.

- Changed accordingly.

Line 236: “on a precision scale”.

- Changed.

Line 238: Delete “for”.

- Done.

Line 268: Technically speaking, feces are not excretion.

- Change to defecation.

Line 292: “from one frame every 20 seconds”.

- Changed.

Line 424: “among body parts”.

- Changed.

Line 426: “were also more similar…”. The rest of the section is written using past tenses, so for consistency’s sake that should be maintained.

- Changed and checked throughout the rest of the section.

Line 427: Remove the comma after “results”.

- Removed.

Line 428: Add a comma before “with a higher…”.

- Comma added.

Line 484: “fail to provide”.

- Changed accordingly.

Line 496: “it will allow”.

- Changed.

Line 499: “difficulties in incorporating”.

- Changed.

Line 511: Delete the comma after “body”.

- Deleted.

Line 520: What is “thermal history”? It should be briefly explained.

- What we mean by “less likely to accumulate thermal history” refers to the cumulative effect of past thermal changes in this body part. In other words, given its small thermal inertia, the environmental temperatures experienced by the animal in the immediate past, as well as its past body temperatures, are less likely to influence the current temperature(s) of the animal/body part. We have now clarified this further in the manuscript.

Line 526: “If this were true”.

- Changed.

Line 561: Add a comma after “Nonetheless”.

- Done.

Line 569: “operative temperature” is a specific concept that refers to the temperature a non-thermoregulating animal would achieve in a given situation, usually resorting to unanimated physical animal models. Is that what you mean here?

- Yes, this is exactly what we mean here. In short, to make rates completely comparable in order to contrast the effect of physiology and/or behaviour on these, the power output of both set-ups should be the same. A way to calibrate this would be to resort to inanimate physical animal models to ensure that the power output of both set-ups would result in the same operative temperatures. Hence, any difference observed in body temperatures and thermal exchange rates would be due to thermoregulatory mechanisms (behavioural, physiological). 

Line 571: The closing parenthesis should be added.

- Added.

Line 582: “testing it on”. This gecko is crepuscular and nocturnal.

- Adjusted accordingly.

Reviewer 2’s Comments:

As the authors indicate, this study introduces a pioneering experimental technique that employs thermography as a less invasive alternative for precisely measuring body temperatures in thermoregulating ectotherms when quantifying thermal exchange rates. This methodology enhances the integration of behaviour and physiology, providing higher temporal and spatial resolution in measuring body temperatures of thermoregulating ectotherms. The study presents findings from original, well-designed research. The experiments and statistical analyses are conducted to a high technical standard and described thoroughly. Therefore, I recommend the article for publication in its current form.

- We thank Reviewer 2 for his/her feedback.

---

## [Editor Report · Decision Letter 1]

9 Dec 2024

Novel method to investigate thermal exchange rates in small, terrestrial ectotherms: A proof-of-concept on the gecko *Tarentola mauritanica*

PONE-D-24-45585R1

Dear Dr. Barroso,

We’re pleased to inform you that your manuscript has been judged scientifically suitable for publication and will be formally accepted for publication once it meets all outstanding technical requirements.

Kind regards,

Edvard Mizsei

Academic Editor

PLOS ONE
---

## [Editor Report · Acceptance letter]

13 Dec 2024

PONE-D-24-45585R1 

PLOS ONE

Dear Dr. Barroso, 

I'm pleased to inform you that your manuscript has been deemed suitable for publication in PLOS ONE. Congratulations! Your manuscript is now being handed over to our production team.

Kind regards, 

on behalf of

Dr. Edvard Mizsei 

Academic Editor

PLOS ONE